# Association between environmental and climatic risk factors and the spatial distribution of cystic and alveolar echinococcosis in Kyrgyzstan

**Giulia Paternoster**[1], **Gianluca Boo**[2☯], **Roman Flury**[3☯], **Kursanbek M. Raimkulov**[4], **Gulnara Minbaeva**[5], **Jumagul Usubalieva**[5], **Maksym Bondarenko**[2], **Beat Müllhaupt**[6], **Peter Deplazes**[7], **Reinhard Furrer**[3,8‡], **Paul R. Torgerson**[1‡*]

**1** Section of Epidemiology, Vetsuisse Faculty, University of Zurich, Zurich, Switzerland, **2** WorldPop, Department of Geography and Environment, University of Southampton, Southampton, United Kingdom, **3** Department of Mathematics, University of Zurich, Zurich, Switzerland, **4** Department of Medical Biology, Genetics and Parasitology, Kyrgyz State Medical Academy named after I.K. Akhunbaev, Bishkek, Kyrgyzstan, **5** Government Sanito-Epidemiology Unit, Ministry of Health of the Kyrgyz Republic, Bishkek, Kyrgyzstan, **6** Clinics of Hepatology and Gastroenterology, University Hospital of Zurich, Zurich, Switzerland, **7** Institute of Parasitology, University of Zurich, Vetsuisse Faculty, Zurich, Switzerland, **8** Department of Computational Science, University of Zurich, Zurich, Switzerland

☯ These authors contributed equally to this work.
‡ RF and PRT also contributed equally to this work.
* paul.torgerson@access.uzh.ch

**Data Availability Statement:** All relevant data are within the manuscript and its Supporting Information files.

## Abstract

### Background

Cystic and alveolar echinococcosis (CE and AE) are neglected tropical diseases caused by *Echinococcus granulosus sensu lato* and *E. multilocularis*, and are emerging zoonoses in Kyrgyzstan. In this country, the spatial distribution of CE and AE surgical incidence in 2014-2016 showed marked heterogeneity across communities, suggesting the presence of ecological determinants underlying CE and AE distributions.

### Methodology/Principal findings

For this reason, in this study we assessed potential associations between community-level confirmed primary CE (no.=2359) or AE (no.=546) cases in 2014-2016 in Kyrgyzstan and environmental and climatic variables derived from satellite-remote sensing datasets using conditional autoregressive models. We also mapped CE and AE relative risk. The number of AE cases was negatively associated with 10-year lag mean annual temperature. Although this time lag should not be considered as an exact measurement but with associated uncertainty, it is consistent with the estimated 10–15-year latency following AE infection. No associations were detected for CE. We also identified several communities at risk for CE or AE where no disease cases were reported in the study period.

### Conclusions/Significance

Our findings support the hypothesis that CE is linked to an anthropogenic cycle and is less affected by environmental risk factors compared to AE, which is believed to result from

**Funding:** PT Swiss National Science Foundation (SNSF, grant agreement number 173131 —"Transmission modelling of emergent echinococcosis in Kyrgyzstan") Funder website: http://www.snf.ch RF Swiss National Science Foundation (SNSF, grant agreement number 175529 - "Disentangling evidence from huge multivariate space-time data from the earth sciences ") Funder website: http://www.snf.ch The funders had no role in study design, data collection and analysis, decision to publish, or preparation of the manuscript.

**Competing interests:** The authors have declared that no competing interests exist.

spillover from a wild life cycle. As CE was not affected by factors we investigated, hence control should not have a geographical focus. In contrast, AE risk areas identified in this study without reported AE cases should be targeted for active disease surveillance in humans. This active surveillance would confirm or exclude AE transmission which might not be reported with the present passive surveillance system. These areas should also be targeted for ecological investigations in the animal hosts.

## Author summary

Cystic and alveolar echinococcosis (CE and AE) are parasitic zoonoses that cause a substantial disease burden in Kyrgyzstan. The etiologic agents of these diseases are parasites in the genus *Echinococcus*. These parasites have complex life cycles which include mammalian definitive and intermediate hosts and a free-living egg stage in the environment. Consequently, environmental and climatic factors can affect the prevalence and geographical distribution of these diseases because such factors influence the parasites' eggs survival and longevity, and can affect suitable habitats for the intermediate and definitive hosts. In this geographic correlation study, we assessed environmental and climatic determinants of the spatial distributions of CE and AE in Kyrgyzstan. We found that 10-year lag annual temperature plays an important role in AE distribution, whilst none of the variables assessed was found to significantly affect that of CE. Moreover, communities at risk where these diseases are potentially under- or misdiagnosed were identified. Our findings provide vital information for targeted, area-specific interventions in Kyrgyzstan, and add to the body of knowledge on the ecology of these neglected parasitic diseases that are emerging and reemerging in several regions in North America, Europe and Asia.

## Introduction

Cystic echinococcosis (CE) and alveolar echinococcosis (AE) are parasitic zoonoses caused by *Echinococcus granulosus sensu lato* and *E. multilocularis*, respectively. *E. granulosus* is generally transmitted in an anthropogenic pastoral cycle between canid definitive host and livestock intermediate host, mainly a dog-sheep-dog cycle. *E. multilocularis* usually occurs in a wildlife cycle involving wild canid (mainly foxes) definitive host and small mammal intermediate hosts. However, dogs can act as AE definitive hosts. Definitive hosts become infected through the ingestion of organs of livestock, hunted game, or small mammals that contain larval cysts called metacestodes. Metacestodes develop into adult tapeworms in the small intestine of definitive hosts, which then shed parasitic eggs into the environment through their feces [1]. Intermediate hosts become infected while grazing, through the ingestion of parasitic eggs that will develop into metacestodes in their organs.

CE and AE infection in humans occurs through the accidental ingestion of parasitic eggs via foodborne, waterborne, or hand-to-mouth transmission routes [2–4]. After a variable latency period (months to years for CE and up to 10-15 years for AE), metacestodes development differs for CE and AE. CE lesions are fluid-filled metacestodes called hydatid cysts that might grow in the liver, lungs, or other organs, causing organ compression. AE lesions are infiltrating cellular protrusions of the metacestode's germinal layer that can metastasize from the liver to neighboring organs. Depending on symptoms and parasitic lesions' characteristics, CE treatment includes a watch-and-wait approach, albendazole treatment, and surgery. AE

treatment includes surgery combined with albendazole treatment or lifelong albendazole treatment [5].

Kyrgyzstan is a mountainous landlocked country located in Central Asia, bordered by Kazakhstan, Uzbekistan, Tajikistan and China. The country has a continental and mostly arid climate [6] covering an area of 199,951 km$^2$. Kyrgyzstan is divided into 7 regions. Bishkek, the capital, and the city of Osh and are administratively equivalent to regions. Further administrative divisions are 56 districts and 479 local communities. The 2016 estimated country's population was $\approx$ 6 million people [7]. In Central Asia, including Kyrgyzstan and its neighbouring countries, both CE and AE appear to be emerging, particularly in the last 30 years after the dissolution of the Soviet Union in 1991. Over 90% of the global burden of AE occurs in neighboring China with an estimated 16000 annual AE cases [2]. Notification of CE and AE confirmed surgical cases to the Government Sanito-Epidemiology Unit in Bishkek is mandatory in Kyrgyzstan. Surgical incidence of CE has increased from 5.4 per 100,000 population per year in 1991 to 13.1 per 100,000 population per year in 2014-2016 [8]. Since the first AE surgical case was recorded in Kyrgyzstan in 1996, an increasing number of cases has been reported, reaching about 180 cases per year (3.02 per 100,000 population per year) in 2014-2016 [8]. The spatial distribution of CE and AE surgical incidence in 2014-2016 in Kyrgyzstan showed remarkable geographic variation across communities. While CE appeared to be widespread, AE was clustered in the southwest and center of the country [8]. This spatial distribution is expected to reflect the presence of ecological risk factors for CE or AE transmission to humans. This is because environmental, topographic, and climatic risk factors, can affect the prevalence and geographical distribution of parasitic zoonoses such as CE and AE by influencing the survival and longevity of parasites' eggs in the environment, exposure to eggs, presence and abundance of competent hosts, their spatial overlap and predation [9–12]. Direct and indirect ecological influences on transmission of *E. granulosus* and *E. multilocularis* are depicted in the conceptual diagram of Atkinson *et al.* [11].

Spatial epidemiology focuses on the study of the relationships between the spatial distribution of diseases and associated risk factors through disease mapping, cluster analysis, and geographic correlation studies [13]. The latter aim at exploring the associations between the spatial distribution of diseases and the exposure to risk factors. Although these studies assess correlations at ecologic or group level and need validation and replication at the individual level, they are useful to generate hypotheses in disease transmission dynamics. Existing evidence supports the association of the environment with the spatial variation of AE, whilst fewer geographic correlation studies have been undertaken to identify environmental determinants of the spatial distribution of CE [10,14]. Despite these links between ecological features and incidence of CE and AE, there are currently no studies evaluating associations between the incidence of these diseases and ecological characteristics in Kyrgyzstan. Consequently, environmental risk factors for *E. granulosus* and *E. multilocularis* infections in humans in Kyrgyzstan are unknown.

We analyzed surgical cases of CE or AE [8] combined with satellite remote sensing data on environmental and climatic factors at the community level in Kyrgyzstan. We used spatial generalized linear mixed models (GLMM) for areal unit data within the conditional autoregressive model (CAR) class [15] and inference in a Bayesian setting, and mapped community-level relative risk for both diseases. Estimated relative risk was defined as the ratio between fitted cases and expected cases at the community level. Our modelling effort aimed at assessing potential associations between the spatial distribution of the CE and AE and ecological determinants, such as temperature or elevation. Our findings are meant to identify high risk areas to inform public health interventions.

## Methods

### Ethics statement

Ethical approval for the study was granted by the ethics committee of the Ministry of Health in Kyrgyzstan.

### Incidence data

As fully described in Paternoster *et al.* [8], we obtained records of confirmed CE and AE surgical cases in Kyrgyzstan between 2014 and 2016 from the Government Sanito-Epidemiology Unit in Bishkek. Each case was manually geocoded according to the patient's community residence, which was assumed to be a proxy for the place of infection. Primary cases (i.e., excluding relapses) were then aggregated at the community-level (no.=490) and linked to the 2009 national census population data to estimate average annual crude surgical incidence of CE and AE at the community level in 2014-2016 [8]. We also computed the number of expected cases of CE and AE using indirect standardization, and standardized incidence ratios (SIR) for both diseases at the community level [8]. Number of CE and AE cases aggregated at different spatial levels are available in the appendix of the work of Paternoster *et al.* [8].

### Explanatory variables preprocessing and selection

Based on previous ecological studies on AE and CE [9–11,14,16–23], we derived 190 geospatial variables on potential environmental and climatic risk factors for CE and AE from different satellite-based remote-sensing data sources. These data sources included the Giovanni online data system of the National Aeronautics and Space Administration (NASA) Goddard Earth Sciences Data and Information Services Center [24], the WorldPop data portal [25,26], Earthstat [27], and WorldClim 2.0 Beta version 1 [28]. All variables' names, description, format, unit, spatial resolution, provider and link are detailed in S1 Table.

To account for the prolonged latency of CE (months to years) and AE (10–15 years) [5], variables were derived for the years 2000, 2005, and 2010, as well as 30-year mean monthly data on temperature (minimum, maximum, and average), solar radiation, and precipitation. We compiled 2016 data for non-temporal variables (i.e., elevation, slope, distance to major waterways and to major roads). Distance to inland waterbodies was available as average for the 2000-2012 period. We also used data on the areal extent and population of communities in 2000, 2005, and 2010 to compute annual population densities (population/km$^2$).

We first harmonized the coordinate reference system of the environmental and climatic data and computed the mean raster values for each community. We then assessed possible linear correlation with the SIR of CE and AE at the community level by using the Spearman's rank correlation coefficient (Spearman's ρ). Nonlinear correlations were also checked by visually inspecting pairwise scatterplots. Variance inflation factor (VIF) was used to detect multicollinearity [29]. We first excluded the variable with greater VIF in any pair of potential explanatory variables with a correlation coefficient ρ>0.9, and then computed all variables' VIF using an iterative stepwise procedure until all correlated variables (VIF>10) were excluded.

We selected variables exhibiting no multicollinearity and significant correlation with the SIR of CE or AE. Other variables were also included based on existing literature (e.g., elevation) [10,16]. We excluded redundant variables (e.g., 2005 and 2010 distance to settlements, keeping the variable for 2000 only). Last, we standardized the selected variables by subtracting the mean and dividing by the standard deviation [16].

## Spatial modeling

We assessed potential associations between selected explanatory variables and community-level cases of CE or AE independently by fitting 6 CAR models (3 for AE and 3 for CE) available in the R package CARBayes [30]. These statistical models belong to the family of GLMM for areal unit data with inference in a Bayesian setting using Markov chain Monte Carlo (MCMC). We previously estimated disease risk for CE and AE based on their SIR [8]. However, for rare diseases or small populations at risk, the SIR can lead to risk under- or overestimation. To overcome this, spatial models estimate risk also using available explanatory variables information and a set of spatial random effects. Spatial random effects are included to account for any overdispersion or spatial autocorrelation in the disease risk after the effect of the exploratory variables have been accounted for [15].

For each disease, we fitted 1) a model with no spatial random effect, i.e., independent noise component only (IND); 2) a Besag, York and Mollié (BYM) model, also called convolution model that contains both a spatially autocorrelated and an independent noise component [31]; 3) a Leroux (LER) model [32], which models spatial autocorrelation based on a single spatial random effect that inherently contains a noise component. For all spatial models, CE and AE community-level cases counts were used as the response variable with a Poisson distribution. To incorporate a disease exposure variable in the count data, we used the natural logarithm of the number of expected CE and AE cases at the community level as a fixed offset. We previously defined the spatial structure of AE and CE incidence data [8] using the contiguity method to capture spatial autocorrelation. As this method is based on a non-negative symmetric neighborhood matrix, communities with no neighbors were excluded from the analysis.

To select the spatial model that best suits the data, we chose the model with lowest deviance information criterion (DIC) value. We also further selected variables based on DIC values. Inference for all spatial models was based on 3 parallel MCMC chains, each of which was run for 300,000 samples, obtained following a burn-in period of 100,000 to achieve convergence and thinning the remaining 200,000 samples by 10 to reduce their autocorrelation. This results in 20,000 samples for inference across the 3 chains. Convergence was assessed through visual inspection of parallel Markov chains' trace plots, and using the Gelman and Rubin's convergence diagnostic to compute multivariate potential scale reduction factor (PSRF), with value<1.1 indicating chain convergence [33]. We tested models' residuals for significant (p<0.05) spatial autocorrelation by using a permutation test (number of permutations =1000) based on the Moran's I statistics [34].

Last, we produced choropleth maps of CE and AE relative risk, defined as the ratio between fitted cases and expected cases at the community level. Relative risk was represented using a combined quantile classification to allow for comparison between the two diseases. We also mapped the posterior exceedance probability (PEP), which is the probability that each community exceeds the average risk of 1, given the data. The PEP ranges between 0 and 1 and provides evidence of significant excess of CE or AE risk within individual areas [15].

All analyses and maps were produced using R software version 3.5.1 [35].

## Results

### Explanatory variables

Variables that exhibited no multicollinearity and had significant correlation with the SIR of CE or AE at the community level were 13 for CE and 17 for AE. We did not detect any nonlinear correlations between the SIR of CE or AE and selected variables by visual inspection of their pairwise scatterplots. Based on literature, we added 3 explanatory variables (i.e., elevation,

proportion of pasture, and 10-year lag mean annual temperature) for CE and 2 explanatory variables (i.e., elevation and 10-year lag mean annual temperature) for AE. Based on DIC values, one variable was removed from AE CAR models. For CE 13 and for AE 16 explanatory variables were selected to fit CAR models (Table 1). For each selected explanatory variable, a descriptive report including, an overview (name and description), metadata (data type, year, unit of measure, spatial resolution, source), summary statistics (mean, standard deviation and range), and a choropleth map with pop-up windows for each community with its relative rounded value is available here: https://www.math.uzh.ch/pages/kgz_eco/.

**Table 1. Selected explanatory variables used in this study in the spatial models for cystic echinococcosis and alveolar echinococcosis in Kyrgyzstan, 2014-2016.**

| Response variable selected for modelling | Variable name | Year | Unit | Spatial resolution | Link |
|---|---|---|---|---|---|
| CE[a] and AE | Distance to built-settlement area edges | 2000 | km | 3" (~ 0.1 km) | https://www.worldpop.org/geodata/summary?id=19149 |
| CE and AE | Distance to herbaceous area edges | 2000 | km | 3" (~ 0.1 km) | https://www.worldpop.org/geodata/summary?id=21639 |
| CE and AE | Distance to woody-tree area edges | 2010 | km | 3" (~ 0.1 km) | https://www.worldpop.org/geodata/summary?id=21639 |
| CE and AE | Elevation[b] | 2000 | m | 3" (~ 0.1 km) | https://www.worldpop.org/geodata/summary?id=23382 |
| CE and AE | Slope | 2000 | degree | 3" (~ 0.1 km) | https://www.worldpop.org/geodata/summary?id=23133 |
| CE and AE | Pasture area fraction[b] | 2000 | area fraction | 5' (~ 10 km) | http://www.earthstat.org/cropland-pasture-area-2000/ |
| CE and AE | Mean winter NDVI[c] | 2010 | NDVI | 0.05˚ (~ 5 km) | https://giovanni.gsfc.nasa.gov/giovanni/ |
| CE and AE | Mean temperature[b] | 2005 | ˚C | 0.5 x 0.625˚ (~ 50 km) | https://giovanni.gsfc.nasa.gov/giovanni/ |
| CE and AE | Mean precipitation in May | 1970-2000 | mm | 30" (~ 1 km) | https://www.worldclim.org/data/worldclim21.html |
| CE | Distance to major waterways | 2016 | km | 3" (~ 0.1 km) | https://www.worldpop.org/geodata/summary?id=17913 |
| CE | Mean spring precipitation | 2000 | mm/h | 0.25˚ (~ 25 km) | https://giovanni.gsfc.nasa.gov/giovanni/ |
| CE | Mean spring precipitation | 2010 | mm/h | 0.25˚ (~ 25 km) | https://giovanni.gsfc.nasa.gov/giovanni/ |
| CE | Mean solar radiation in February | 1970-2000 | kJ m$^{-2}$ day$^{-1}$ | 30" (~ 1 km) | https://www.worldclim.org/data/worldclim21.html |
| AE | Distance to cultivated area edges | 2010 | km | 3" (~ 0.1 km) | https://www.worldpop.org/geodata/summary?id=21639 |
| AE | Distance to urban artificial area edges | 2000 | km | 3" (~ 0.1 km) | https://www.worldpop.org/geodata/summary?id=21639 |
| AE | Night-time lights VRIIS[d] | 2015 | nanowatts cm$^{-2}$ sr$^{-1}$ | 3" (~ 0.1 km) | https://www.worldpop.org/geodata/summary?id=18651 |
| AE | Mean winter precipitation | 2000 | mm/h | 0.25˚ (~ 25 km) | https://giovanni.gsfc.nasa.gov/giovanni/ |
| AE | Mean winter precipitation | 2005 | mm/h | 0.25˚ (~ 25 km) | https://giovanni.gsfc.nasa.gov/giovanni/ |
| AE | Mean solar radiation in November | 1970-2000 | kJ m$^{-2}$ day$^{-1}$ | 30" (~ 1 km) | https://www.worldclim.org/data/worldclim21.html |
| AE | Population density | 2000 | Population km$^{-2}$ | Community | https://www.worldpop.org/geodata/listing?id=29 |

[a]CE cystic echinococcosis, AE alveolar echinococcosis

[b]Variable added from literature

[c]NDVI: Normalized Difference Vegetation Index

[d]VRIIS: Visible Infrared Imaging Radiometer Suite

Last access links: 13/08/2020

**Table 2. Deviance information criterion value of six types of conditional autoregressive models for cystic echinococcosis and alveolar echinococcosis in Kyrgyzstan, 2014-2016.**

| | Independent model | Besag, York and Mollié model | Leroux model |
|---|---|---|---|
| | DIC[a] value | DIC* value | DIC* value |
| Cystic echinococcosis | 2814.73767 | 2075.0901 | 2096.1816 |
| Alveolar echinococcosis | 1461.773 | 972.4378 | 986.1978 |

[a]DIC: Deviance Information Criterion.

## Spatial modelling

Of the 490 communities of Kyrgyzstan, 12 did not have neighboring communities [8] and were thus excluded from spatial models for a total of 478 spatial units of analysis. Kyrgyz communities have the peculiarity of not covering the entire country, but only the settlements and their surroundings. Twelve communities in this country do not border on other communities. Based on DIC estimates, CAR BYM models had best model fit (lower DIC value) for CE (DIC = 2075.52) and AE (DIC = 972.44) among all the models examined (Table 2). The higher DIC values for the IND models indicate that for both diseases, the model fit was improved by additional spatial random effect. PSRF was 1.05 for CE and 1.07 for AE, indicating the convergence of the three parallel MCMC chains.

Posterior medians and 95% credible intervals (CI) of BYM models' combined MCMC chains are summarized in Table 3 for CE and in Table 4 for AE. For CE, none of the variables considered in the model exhibited significant effect on the number of CE cases as their 95% CI included 0. By contrast, we found a significant negative association between mean annual temperature in 2005 and AE cases (posterior median -0.62, 95% CI -1.17 to -0.04). This implies an estimated decrease of 46.2% (CI -69.01 to -3.49) in AE number of cases for a 1°C increase in

**Table 3. Variables, regression coefficients (posterior medians), relative risk, 95% credible intervals from conditional autoregressive Besag, York and Mollié model for cystic echinococcosis in Kyrgyzstan, 2014-2016.**

| Model | Conditional autoregressive Besag, York and Mollié model for cystic echinococcosis, deviance information criterion = 2075.0901 | | | |
|---|---|---|---|---|
| Variable | Coefficient, posterior median | (95% CI[a]) | Relative risk, posterior median | (95% CI) |
| Intercept | 0.08 | (-0.19 - 0.34) | - | |
| Mean winter[b] NDVI[c] 2010 | -0.10 | (-0.32 - 0.13) | 0.90 | (0.72 - 1.14) |
| Mean spring[b] precipitation 2000 | -0.09 | (-0.31-0.12) | 0.92 | (0.73 - 1.13) |
| Mean spring[b] precipitation 2010 | 0.21 | (-0.08-0.50) | 1.23 | (0.92 - 1.65) |
| Distance to woody-tree area edges 2010 | -0.01 | (-0.15-0.12) | 0.99 | (0.86 - 1.13) |
| Distance to herbaceous area edges 2000 | 0.08 | (-0.03-0.20) | 1.09 | (0.97 - 1.23) |
| Distance to built-settlement area edges 2000 | -0.03 | (-0.23-0.16) | 0.97 | (0.8 - 1.18) |
| Distance to major waterways 2016 | 0.03 | (-0.07-0.14) | 1.03 | (0.93 - 1.15) |
| Slope 2000 | 0.02 | (-0.17-0.21) | 1.02 | (0.85 - 1.24) |
| Mean precipitation in May 1970-2000 | -0.03 | (-0.24-0.17) | 0.97 | (0.78 - 1.19) |
| Mean solar radiation in February 1970-2000 | -0.01 | (-0.29-0.26) | 0.99 | (0.75 - 1.3) |
| Elevation 2000 | 0.09 | (-0.27-0.45) | 1.09 | (0.76 - 1.57) |
| Pasture area fraction 2000 | 0.02 | (-0.11-0.14) | 1.02 | (0.9 - 1.15) |
| Mean temperature 2005 | -0.12 | (-0.36-0.10) | 0.88 | (0.7 - 1.11) |

[a]CI: credible interval

[b]winter: from December through February; spring: from March through May

[c]NDVI: Normalized Difference Vegetation Index

**Table 4. Variables, regression coefficients (posterior medians), relative risk, 95% credible intervals from conditional autoregressive Besag, York and Mollié or convolution model for alveolar echinococcosis in Kyrgyzstan, 2014-2016.**

| Model | Conditional autoregressive Besag, York and Mollie or convolution model for alveolar echinococcosis, deviance information criterion = 972.4378 | | | |
|---|---|---|---|---|
| Variable | Coefficient posterior median | (95% CI[a]) | Relative risk, posterior median | (95% CI) |
| Intercept | -1.28 | (-1.98 to -0.7) | - | |
| Mean winter[b] NDVI[c] 2010 | 0.28 | (-0.24 - 0.8) | 1.32 | (0.78 - 2.23) |
| Mean winter[b] precipitation 2000 | -0.45 | (-1.1 - 0.17) | 0.64 | (0.33 - 1.18) |
| Mean winter[b] precipitation 2005 | 0.23 | (-0.58 - 1.07) | 1.26 | (0.56 - 2.93) |
| Pasture area fraction 2000 | 0.01 | (-0.25 - 0.28) | 1.01 | (0.78 - 1.32) |
| Distance to cultivated area edges 2010 | 0.09 | (-0.27 - 0.43) | 1.09 | (0.76 - 1.53) |
| Distance to woody-tree area edges 2010 | -0.18 | (-0.5 - 0.14) | 0.83 | (0.6 - 1.15) |
| Distance to herbaceous area edges 2000 | -0.14 | (-0.47 - 0.17) | 0.87 | (0.62 - 1.19) |
| Distance to urban artificial area edges 2000 | -0.02 | (-0.53 - 0.52) | 0.98 | (0.59 - 1.68) |
| Night-time lights VRIIS[d] 2015 | -0.11 | (-0.35 - 0.15) | 0.9 | (0.7 - 1.16) |
| Distance to built-settlement area edges 2000 | -0.27 | (-0.64 - 0.11) | 0.77 | (0.53 - 1.12) |
| Slope 2000 | -0.28 | (-0.69 - 0.17) | 0.75 | (0.5 - 1.19) |
| Mean precipitation in May 1970-2000 | 0.28 | (-0.17 - 0.75) | 1.33 | (0.84 - 2.12) |
| Mean solar radiation in November 1970-2000 | 0.56 | (-0.09 - 1.25) | 1.74 | (0.91 - 3.49) |
| Population density 2000 | -0.09 | (-0.21 - 0.03) | 0.91 | (0.81 - 1.03) |
| Elevation 2000 | 0.82 | (-0.05 - 1.63) | 2.26 | (0.95 - 5.11) |
| **Mean temperature 2005** | **-0.62** | **(-1.17 to -0.04)** | **0.54** | **(0.31 - 0.97)** |

[a]CI: credible interval

[b]winter: from December through February

[c]NDVI: Normalized Difference Vegetation Index

[d]VRIIS: Visible Infrared Imaging Radiometer Suite

Boldface indicates significant results, i.e., posterior CI for median does not contain 0.

annual mean temperature 10 years prior to the diagnosis of the infection (RR = 0.54, 95% CI 0.31-0.97).

Thirteen variables among the 190 variables considered initially (S1 Table) were excluded from the BYM model for AE because they exhibited collinearity with the mean 10-year lag temperature (*i.e.*, the significant predictor for AE). These variables were seasonal (four seasons) temperature in 2005, 2010, 2000, and mean annual temperature in 2000 (S2 Table).

BYM models accounted for residual spatial autocorrelation because we did not detect significant spatial autocorrelation in the residuals of CE BYM model (Moran's I = -0.09, p>0.99) and of AE BYM model (Moran's I = -0.04, p = 0.88).

Estimated relative risk (RR) for CE (Fig 1) was distributed across communities in the whole country and ranged between 0 and 7.72. By contrast, estimated RR for AE (Fig 2) ranged between 0 and 81 and was concentrated in the eastern and central part of the country. Our AE and CE risk maps accounting for environmental and climatic risk factors identified a number of communities at risk where no cases had been reported in the study period (S3 and S4 Tables). PEP maps are shown in S1 Fig for CE and S2 Fig for AE.

## Discussion

This study aimed to describe potential associations between the number of community-level CE or AE cases with environmental and climatic risk factors in Kyrgyzstan. Although the life cycles of *E. granulosus* and *E. multilocularis* are associated with particular environmental and

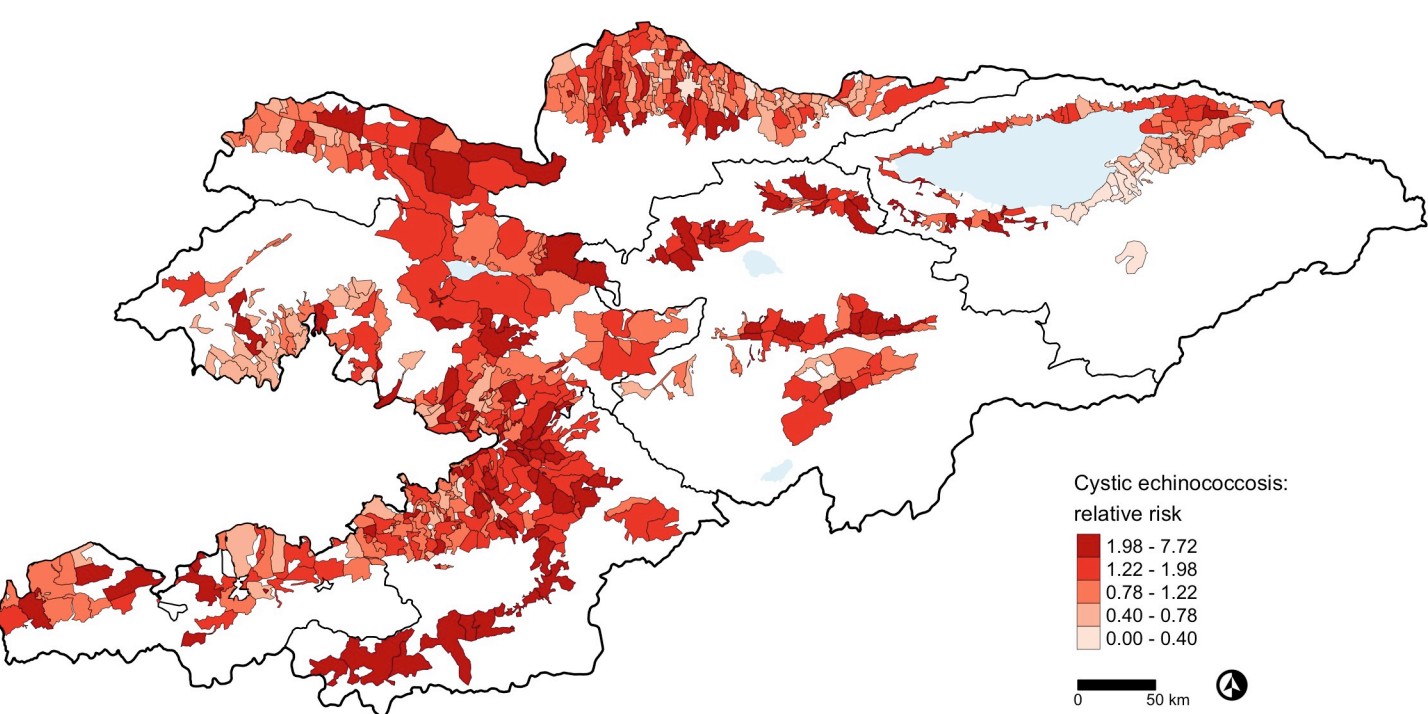

**Fig 1. Estimated relative risk based on the Besag, York and Mollié model for cystic echinococcosis in Kyrgyzstan, 2014-2016.** The local administrative units of Kyrgyzstan cover only the settlements and their surroundings. The white spaces are unsettled areas that do not fall within any local administrative unit. We used the third level administrative boundaries of Kyrgyzstan as provided by REACH, a joint initiative of IMPACT, ACTED, and the UN Operational Satellite Applications Programme (UNOSAT) under a humanitarian license. The shapefiles were subsequently edited to add missing polygons. The code and files for the map are available at: https://git. math.uzh.ch/reinhard.furrer/Echin_kgz. Shape files for third level administrative regions are now freely available from the United Nations Office for the Coordination of Humanitarian Affairs https://data.humdata.org/dataset/kyrgyzstan-administrative-boundaries.

climatic features which have been identified as risk factors for CE and AE transmission [10], these factors have never been previously evaluated in Kyrgyzstan. In this study, we found a negative association between mean annual 10-year lag temperature and the number of AE cases at the community level. None of the selected explanatory variables were found to be associated with the number of community-level CE cases. We also estimated the relative risk for both diseases accounting for environmental and climatic variables, and detected communities at risk of CE or AE where no disease cases had been reported in the study period (S2 and S3 Tables). These communities might be areas of Kyrgyzstan where CE or AE are under- or misdiagnosed.

The 10-year lag annual mean temperature was negatively associated with the number of AE cases in the BYM model, confirming the link between AE and climate [12]. This association potentially reflects the role of temperature on the survival and longevity of *E. multilocularis* eggs in the environment [1,22], in line with another study in China [21]. Eggs of *E. multilocularis* are resistant to freezing but sensible to warmth and desiccation; they can survive for 478 days at 4˚C and 95% relative humidity, but for one day only at 25˚C and 27% relative humidity [22]. *E. granulosus* eggs remain viable for months and resist a range of temperatures between –30˚C to +30˚C [12]. Although there is not experimental evidence that eggs of both *Echinococcus* species have strongly different temperature ranges, eggs survival might have a role explaining why AE only occurs in the northern hemisphere whilst CE has a worldwide distribution [2], and why AE was clustered in the eastern and colder part of Kyrgyzstan (https://www.math.uzh.ch/pages/kgz_eco/).

Temperature might act as a greater constraint on transmission of AE compared to that of CE also because of its influence on AE definitive and intermediate host populations' dynamics

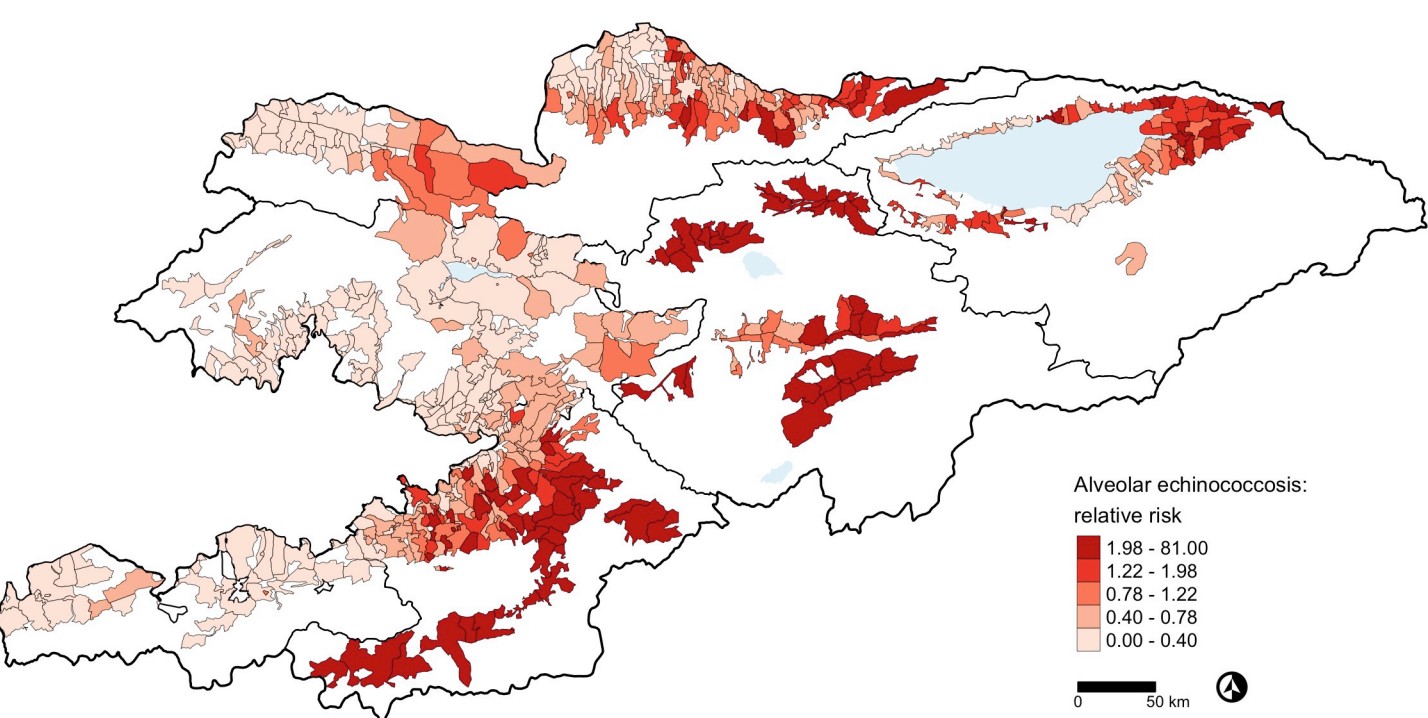

**Fig 2. Estimated relative risk based on the Besag, York and Mollié model for alveolar echinococcosis in Kyrgyzstan, 2014-2016.** The local administrative units of Kyrgyzstan cover only the settlements and their surroundings. The white spaces are unsettled areas that do not fall within any local administrative unit. We used the third level administrative boundaries of Kyrgyzstan as provided by REACH, a joint initiative of IMPACT, ACTED, and the UN Operational Satellite Applications Programme (UNOSAT) under a humanitarian license. The shapefiles were subsequently edited to add missing polygons. The code and files for the map are available at: https://git. math.uzh.ch/reinhard.furrer/Echin_kgz. Shape files for third level administrative regions are now freely available from the United Nations Office for the Coordination of Humanitarian Affairs https://data.humdata.org/dataset/kyrgyzstan-administrative-boundaries.

(i.e., foxes and small mammals and predator-prey relationships between foxes, dogs, and small mammals) [21,36]. By contrast, CE occurs in an anthropogenic pastoral cycle where human exposure to *E. granulosus* eggs is facilitated by frequent dog-sheep-dog-humans contacts maintained by human activities such as livestock rearing, home slaughtering, and feeding infected offal to dogs. Although home slaughtering of livestock is prohibited by the law in Kyrgyzstan, it is common in rural areas [37]. Multicollinearity detected between the mean 10-year lag temperature and other mean temperature variables (S2 Table) can be expected when a variable computed in different time points displays limited temporal variability or strong spatial patterns. In our study, it is possible that the stability or strong spatial patterns of temperature in Kyrgyzstan might be due to the topography and landscapes (e.g., rugged relief with valleys) that play an important role in defining climatic regions.

None of the 13 selected environmental and climatic factors were associated with the number of community-level CE cases in the study period in the BYM model (Table 3), and CE relative risk was widespread (Fig 1). This suggests that there might be other spatially explicit unmeasured factors impacting CE distribution and risk in Kyrgyzstan, similarly to another study in China [27]. Given the pastoral cycle of CE, these unmeasured factors likely entail socio-economic and behavioral factors (e.g. poverty, dog ownership, home slaughtering). Average 10-year lag winter and 13-year lag annual temperature were reported to be associated with CE incidence in the Ningxia Hui Autonomous Region in China [21] but accounted for a relatively small proportion of the spatiotemporal variation in CE risk in this area.

Neither distance to cultivated, woody-tree, herbaceous area edges nor NDVI appeared to be associated with AE cases in the BYM model (Table 4). This contrasts other studies that

identified the presence of specific landcover or vegetation types as a risk factor for AE as they provide a habitat supporting abundant presence of suitable small mammal species [12]. A high proportion of grassland was found to be associated with AE in humans in China [9,23] and Central Asia [38], whilst the area of cultivated land was found to have a negative correlation with AE [23]. Although population density of AE intermediate hosts was linked with grassland productivity in southern Kyrgyzstan [14], in our study it is possible that temperature had a bigger impact in influencing the presence and abundance of AE suitable intermediate hosts.

The main limitation of this study is the use of CE and AE surgical cases as response variable in our models. Although CE and AE surgical cases are notifiable in Kyrgyzstan, they represent only a proportion of cases [39] as both diseases can be asymptomatic for prolonged periods, can be managed with approaches different than surgery, and are often misdiagnosed or under-reported. This might bias our results by underestimating or overestimating the association between environmental and climatic variables and the number of disease cases. For example, the association between the mean 10-year lag temperature and the number of AE cases could be underestimated if the distribution of surgical cases and that of less advanced or AE cases would be clustered in the same way, or overestimated, should the less advanced or non-diagnosed AE cases be widespread across the country. Active surveillance (*e.g.*, ultrasound surveys) throughout Kyrgyzstan could provide data on the number and location of less advanced disease cases. However, such population-based imaging studies are usually done in limited areas of known high endemicity, such as the study in the village of Sary Mogol in southern Kyrgyzstan [39]. This village was identified as an AE cluster based on hospital records of AE surgical cases. The ultrasound survey performed in 2012 on 1617 persons confirmed the cluster uncovering a high prevalence of 4.2% probable or confirmed AE cases. A cross-sectional ultrasound-based survey based on a large sample size (over 24000 volunteers from 50 villages) was done in rural Bulgaria, Romania, and Turkey to estimate CE prevalence in Eastern Europe [4]. This study found active CE cysts in people of all ages and in all investigated villages. These villages were located in provinces with annual CE hospital incidence within the mid-range for the respective countries. Thus, we can assume that also in Kyrgyzstan the distribution of severe and less severe CE cases might be similar, minimizing possible biases.

Financial constraints preventing access to diagnosis and treatment might influence the number of diagnosed surgical cases and their location. However, previous analyses on the same data set showed no significant correlation between the location of the reported AE or CE surgical cases and the number or distance to the closest health facility in Kyrgyz local communities [8]. This provides evidence that possible bias is minimized. Finally, the size of our data set (2359 cases of CE and 546 cases of AE from a short time period of 3 years) adds robustness to our analyses, even if non-surgical cases could not be analyzed.

This study was also limited by the environmental and climatic data used, which were obtained at a 5-, 10-, and 15-year lag and in monthly averages over a 30-year period to account for the variable long incubation period of both diseases, similarly to the work of Cadavid Restrepo *et al.* [21]. However, the moment of infection with *Echinococcus spp.* is not traceable in space and time [4]. Moreover, the correlation of temperature data from different years (S2 Table) shows a stability of this variable over time and space. Thus, the exact time lag between the measured temperature data and the associated higher number of cases cannot be precisely established from this dataset and should be considered with a degree of uncertainty. Although AE intermediate hosts have specific habitat requirements and thus landscape composition is considered as a proxy for their distribution [14], the lack of adequate data on AE intermediate hosts distribution and abundance also limits our study. Additionally, it remains unknown whether the emergence of AE in Kyrgyzstan might also be due to a higher virulence of the

most common variant infecting humans in this country [40]. More data and genotyping of both human and definitive host populations might help understanding this issue.

Future studies would benefit from including high-resolution socioeconomic and behavioral factors for CE and AE. These risk factors include the presence of free roaming dogs, their feeding with viscera, home slaughtering, living in rural areas and low income for CE; hunting foxes, dog ownership, low income, low education, and source of drinking water for AE [41].

Our spatial analyses of ecologic determinants of CE and AE distribution allowed to produce relative risk maps that can help targeting public health interventions in communities where CE and AE might be under- or misdiagnosed. These communities could be targeted by area-specific health education (e.g., early disease detection and treatment, prevention in the animal hosts, information campaigns for the general population) and active surveillance (e.g. ultrasound survey in the population). To identify socioeconomic and behavioural risk factors for CE and AE in Kyrgyzstan additional population-based studies are needed. Nevertheless, this study indicates that temperature plays an important explanatory role in determining the AE spatial distribution. Therefore climate change may affect this distribution and spatial modelling could be used to predict where the risk of occurrence will be in response to future rising temperatures.

## Supporting information

**S1 Fig. Posterior exceedance probability (PEP) based on the Besag, York and Mollié model for cystic echinococcosis echinococcosis in Kyrgyzstan, 2014-2016.** The PEP ranges between 0 and 1 and provides evidence of significant excess of CE or AE risk within individual areas. We used the third level administrative boundaries of Kyrgyzstan as provided by REACH, a joint initiative of IMPACT, ACTED, and the UN Operational Satellite Applications Programme (UNOSAT) under a humanitarian license. The shapefiles were subsequently edited to add missing polygons. The code and files for the map are available at: https://git.math.uzh.ch/reinhard.furrer/Echin_kgz. Shape files for third level administrative regions are now freely available from the United Nations Office for the Coordination of Humanitarian Affairs https://data.humdata.org/dataset/kyrgyzstan-administrative-boundaries.
(TIF)

**S2 Fig. Posterior exceedance probability (PEP) based on the Besag, York and Mollié model for cystic alveolar echinococcosis in Kyrgyzstan, 2014-2016.** The PEP ranges between 0 and 1 and provides evidence of significant excess of CE or AE risk within individual areas. We used the third level administrative boundaries of Kyrgyzstan as provided by REACH, a joint initiative of IMPACT, ACTED, and the UN Operational Satellite Applications Programme (UNOSAT) under a humanitarian license. The shapefiles were subsequently edited to add missing polygons. The code and files for the map are available at: https://git.math.uzh.ch/reinhard.furrer/Echin_kgz. Shape files for third level administrative regions are now freely available from the United Nations Office for the Coordination of Humanitarian Affairs https://data.humdata.org/dataset/kyrgyzstan-administrative-boundaries.
(TIF)

**S1 Table. Geospatial variables (no. 190) on potential environmental and climatic risk factors for cystic echinococcosis and alveolar echinococcosis in Kyrgyzstan collected for this analysis.**
(DOC)

**S2 Table. Variables (no. 13) with significant correlation (p<0.05) with mean annual temperature in 2005 in Kyrgyzstan.**
(DOC)

**S3 Table. Communities where no cystic echinococcosis cases where reported in the study period that are at risk for cystic echinococcosis (relative risk higher than 1) according to the Besag, York and Mollié model, Kyrgyzstan, 2014-2016.**
(DOC)

**S4 Table. Communities where no alveolar echinococcosis cases where reported in the study period that are at risk for alveolar echinococcosis (relative risk higher than 1) according to the Besag, York and Mollié model, Kyrgyzstan, 2014-2016.**
(DOC)

## Acknowledgments

We acknowledge Dr. Katharina Stärk for her supervision of this research.

## Author Contributions

**Conceptualization:** Giulia Paternoster, Paul R. Torgerson.

**Data curation:** Giulia Paternoster, Gianluca Boo, Roman Flury, Kursanbek M. Raimkulov, Gulnara Minbaeva, Jumagul Usubalieva, Maksym Bondarenko, Reinhard Furrer.

**Formal analysis:** Giulia Paternoster, Gianluca Boo, Roman Flury, Maksym Bondarenko, Reinhard Furrer.

**Funding acquisition:** Giulia Paternoster, Beat Müllhaupt, Peter Deplazes, Reinhard Furrer, Paul R. Torgerson.

**Investigation:** Giulia Paternoster, Kursanbek M. Raimkulov, Gulnara Minbaeva, Maksym Bondarenko, Beat Müllhaupt, Peter Deplazes, Reinhard Furrer, Paul R. Torgerson.

**Methodology:** Giulia Paternoster, Gianluca Boo, Roman Flury, Maksym Bondarenko, Reinhard Furrer, Paul R. Torgerson.

**Project administration:** Gulnara Minbaeva, Jumagul Usubalieva, Peter Deplazes, Paul R. Torgerson.

**Resources:** Kursanbek M. Raimkulov, Gulnara Minbaeva, Jumagul Usubalieva, Maksym Bondarenko, Peter Deplazes, Reinhard Furrer, Paul R. Torgerson.

**Software:** Maksym Bondarenko, Reinhard Furrer.

**Supervision:** Giulia Paternoster, Paul R. Torgerson.

**Validation:** Giulia Paternoster, Maksym Bondarenko, Paul R. Torgerson.

**Visualization:** Giulia Paternoster, Paul R. Torgerson.

**Writing – original draft:** Giulia Paternoster, Gianluca Boo, Roman Flury, Maksym Bondarenko, Beat Müllhaupt, Peter Deplazes, Reinhard Furrer, Paul R. Torgerson.

**Writing – review & editing:** Giulia Paternoster, Peter Deplazes, Reinhard Furrer, Paul R. Torgerson.

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
