## [Decision Letter · Decision Letter 0]

10 Dec 2020

Dear Prof Torgerson,

Thank you very much for submitting your manuscript "Association between environmental and climatic risk factors and the spatial distribution of cystic and alveolar echinococcosis in Kyrgyzstan" for consideration at PLOS Neglected Tropical Diseases. As with all papers reviewed by the journal, your manuscript was reviewed by members of the editorial board and by several independent reviewers. In light of the reviews (below this email), we would like to invite the resubmission of a significantly-revised version that takes into account the reviewers' comments. 

Please address all reviewer concerns in an updated manuscript.

Also, take note to cite all sources using PLOS NTD approved citation styles (including websites and new media).

In addition to addressing concerns raised by the reviewers, please consider to expand your explanation in the methods and table S1 of which variables were excluded due to collinearity - and try to explain for each excluded variable: excluded due to collinearity with which retained variables? In particular, for the significant predictor for AE of 10-year lagged temperature, were there other variables that were excluded that were highly collinear with 10-year lagged temperature? If so, consider to be explicit about those and add speculation in the discussion as to the potential mechanisms involving the other excluded variables that were also collinear with 10-year lagged temperature (if any).

I see some discrepancy between reviewers' concerns that limitations were addressed (mainly the limitation of using surgical cases as a proxy for all cases). I agree with reviewer 3 that more explanation of how focusing on surgical cases might limit interpretations is warranted (rather than simply a statement that surgical cases limit interpretation - in what ways exactly would this bias the results)?

Consider to add a lifecycle figure for each parasite, with the lifecycle figures enhanced to show where environmental variables are hypothesized to influence transmission. This basic mechanistic information about the potential direct and indirect influences of environmental variables on CE and AE transmission is hard to glean from the current manuscript as written.

Line 285 references a Table 5 but I do not see a table 5 in the manuscript - please rectify?

Finally, when you refer to relative risk throughout the manuscript, it is unclear relative to what? Please state your definition of relative risk explicitly.

We cannot make any decision about publication until we have seen the revised manuscript and your response to the reviewers' comments. Your revised manuscript is also likely to be sent to reviewers for further evaluation.

Sincerely,

Susanne H. Sokolow

Associate Editor

Justin Remais

Deputy Editor

Please address all reviewer concerns in an updated manuscript.

Also, take note to cite all sources using PLOS NTD approved citation styles (including websites and new media).

In addition to addressing concerns raised by the reviewers, please consider to expand your explanation in the methods and table S1 of which variables were excluded due to collinearity - and try to explain for each excluded variable: excluded due to collinearity with which retained variables? In particular, for the significant predictor for AE of 10-year lagged temperature, were there other variables that were excluded that were highly collinear with 10-year lagged temperature? If so, consider to be explicit about those and add speculation in the discussion as to the potential mechanisms involving the other excluded variables that were also collinear with 10-year lagged temperature (if any).

I see some discrepancy between reviewers' concerns that limitations were addressed (mainly the limitation of using surgical cases as a proxy for all cases). I agree with reviewer 3 that more explanation of how focusing on surgical cases might limit interpretations is warranted (rather than simply a statement that surgical cases limit interpretation - in what ways exactly would this bias the results)?

Consider to add a lifecycle figure for each parasite, with the lifecycle figures enhanced to show where environmental variables are hypothesized to influence transmission. This basic mechanistic information about the potential direct and indirect influences of environmental variables on CE and AE transmission is hard to glean from the current manuscript as written.

Line 285 references a Table 5 but I do not see a table 5 in the manuscript - please rectify?

Finally, when you refer to relative risk throughout the manuscript, it is unclear relative to what? Please state your definition of relative risk explicitly.

Reviewer's Responses to Questions

**Key Review Criteria Required for Acceptance?**

**Methods**

-Are the objectives of the study clearly articulated with a clear testable hypothesis stated?

-Is the study design appropriate to address the stated objectives?

-Is the population clearly described and appropriate for the hypothesis being tested?

-Is the sample size sufficient to ensure adequate power to address the hypothesis being tested?

-Were correct statistical analysis used to support conclusions?

-Are there concerns about ethical or regulatory requirements being met?

Reviewer #1: Yes

Reviewer #2: -Are the objectives of the study clearly articulated with a clear testable hypothesis stated?

Yes.

-Is the study design appropriate to address the stated objectives?

Yes, with some limitations that, nevertheless, are clearly presented and discussed in the manuscript

-Is the population clearly described and appropriate for the hypothesis being tested?

Yes.

-Is the sample size sufficient to ensure adequate power to address the hypothesis being tested?

Unfortunately, Authors could only use surgery cases for AE and CE as response variable, instead of true notified cases (most likely not-reported/under-reported/misdiagnosed). However, I agree that these data were actually the best available option, so I won’t stress this point as a major flaw of their study at all.

-Were correct statistical analysis used to support conclusions?

Yes.

-Are there concerns about ethical or regulatory requirements being met?

No concerns.

Reviewer #3: The objectives of the study should be more clearly stated in the last paragraph of the introduction. As you are only studying surgical cases, which represent a small subset of total prevalent cases, the inherent bias in this study design should be clearly and extensively addressed in the discussion. Statistical methodologies and modeling were sound and well described. No concerns with ethics. Recommend disclosing the approved protocol number for transparency. It would be nice to have a supplemental table with the numbers of surgical incidence cases (primary cases that were aggregated) by geographic area/unit. It would improve understanding to have additional specific information about where the initial 190 geospatial variables were derived from (previous literature? other?).

**Results**

-Does the analysis presented match the analysis plan?

-Are the results clearly and completely presented?

-Are the figures (Tables, Images) of sufficient quality for clarity?

Reviewer #1: Results are not well described. Part of the results section still look like methodology. Results are not completely presented.

Reviewer #2: -Does the analysis presented match the analysis plan?

Yes.

-Are the results clearly and completely presented?

Yes.

-Are the figures (Tables, Images) of sufficient quality for clarity?

Yes. Maybe a physical landscape map (graphical abstract?) could help the reader that is not familiar with Kyrgyzstan to understand the Country’s territory?

Reviewer #3: The analysis and results are clearly presented and match the methodology. Tables are clear and concise. Figures are nicely presented and clear.

**Conclusions**

-Are the conclusions supported by the data presented?

-Are the limitations of analysis clearly described?

-Do the authors discuss how these data can be helpful to advance our understanding of the topic under study?

-Is public health relevance addressed?

Reviewer #1: Yes

Reviewer #2: -Are the conclusions supported by the data presented? Yes.

-Are the limitations of analysis clearly described? Absolutely.

-Do the authors discuss how these data can be helpful to advance our understanding of the topic under study? Yes.

-Is public health relevance addressed? Yes.

Reviewer #3: Although the discussion of the explanatory variables is thorough, the limitations of the study group - surgical cases - need to be emphasized. Surgical incidence is not a strong proxy for transmission, and should be cautiously and thoughtfully used a proxy for disease incidence. Additional sentences in the discussion about potential implications of this limitation should be added. Although you do touch on the utility of socioeconomic factors to be considered in disease distribution, it is worth adding that these factors influence healthcare access to reach diagnosis and/or surgery in cases.

**Editorial and Data Presentation Modifications?**

Reviewer #1: Not applicable

Reviewer #2: ABSTRACT

- Line 36: I disagree with AE and CE being called “emerging” zoonoses. Is that the case for Kyrgyzstan? Or is it rather an endemic region? please clarify your statement or either just rename them properly as “zoonotic parasitic infections” or “neglected tropical diseases”

- Line 46: 10-year lag?

INTRODUCTION

- LINE 74: same as for line 36. Pls check all through the manuscript

- Line 95: is there some infection prevalence data for your Country / neighbouring areas?

- Line 116: spatial variation OF AE

M&M

Line 132: paragraph “3.1 Incidence data” mostly describes what authors did in a previous publication (reference 8). Please consider to shorten this part, since methods were fully described elsewhere.

Line 139: please clarify “independent cities

Line 213: please check here (www.r-bloggers.com/2018/06/its-easy-to-cite-and-reference-r/) or here (https://cran.r-project.org/doc/FAQ/R-FAQ.html) how to correctly cite R software and its packages. It should go as: (R Core Team, YEAR), with YEAR = release year of the version of R you used.

RESULTS

Lines 222-224: “For both diseases, we removed distance to settlements in 2005 and 2010 because redundant with distance to settlements in 2000.” Authors already stated this in M&M. please delete from here to avoid redundancy.

Line 226: please rephrase as “For each selected explanatory variable, a descriptive report including (…)”

Table 1: First column – maybe “response variable” selected for modelling fits better..?

Line 239: “12 did not have neighbors” please provide better explanation

Line 242: I think “The higher DIC values for the IND models indicate that for both diseases, the model fit was improved by additional spatial random effect” is somehow clearer

Lines 285 & 290: I couldn’t open the url

DISCUSSION

Line 327: Similarly

Reviewer #3: See editorial comments by line below:

Line 87 - this sentence is unclear; consider rewording to "....AE) metacestode development differs for CE and AE" or similar

Line 88 - recommend changing "fluid-filled metacestodes" to hydatid cysts

Line 89,90 - AE lesions are really infiltrative proliferations of metcestode cells....consider rewording protrusions

Line 94 - change "landlock" to "landlocked"; "sharp continental" unclear intent in this wording

Line 97 to 98- notifiable to what agency?

Lines 110 - 115 may be removed beginning at paragraph start and ending at "hypotheses in disease transmission dynamics"

Line 130 - provide approved protocol number if available

Line 133 - use authors name et al. in "As described in..."

Line 136 - replace "surrogate" with "proxy" (also, not that place of residence may not likely represent place of infection, especially in communities where farmers travel for work, etc.)

Line 293 - remove "potential"; the word potential is used twice in that sentence

Line 345 - change to "are often misdiagnosed and underreported"

Line 349 - change "infectious moment" to "moment of infection"

Line 357 - remove hyphen from socio-economic (socioeconomic)

Lines 367 to 371 - reword these for clarity and eloquence

**Summary and General Comments**

Reviewer #1: This is a great research describing potential association between environmental/climatic risk factors and the spatial distribution of cystic and alveolar echinococcosis. It can be further revised to generally address some long winding (and perhaps unnecessary) statements. Results are not well described. Any part of the results that has to do with 'how the work was performed' should be taken to Methods. Likewise, any part of the results that is describing implication of the result should be transferred to Discussion. Please state or describe your results in detail under Results. Give implications of the study or results in Discussion section.

Please minimize the use of the word "we", it makes the paper boring - you may rather use third person reporting format in most cases.

In line 95, did you mean Kyrgyzstan is divided in 7 regions, 2 main cities ... or into 7 regions, ...?

Reviewer #2: The manuscript is generally well written and interesting. A high degree of detail was provided for Methods and also the study’s current main limitations (e.g. AE and CE surgery cases as response variable) were clearly presented and discussed. I really appreciated that.

The study might suffer from a limited geographical significance but, nevertheless, spatial epidemiology methods are certainly worth to be published & shared for further applications in the NTDs environmental modelling field.

I feel like the manuscript is actually suitable for publication thank to its M&M, after minor revision, on the one condition that the Editor finds their country-level “application” suitable for high-ranking PLOS-ntd.

Reviewer #3: This paper presents sound and interesting methodologies and the data is well presented; however, there is concern for interpretation of the findings due to only the inclusion of surgical cases of CE/AE. The implications of using only surgical cases and the limitations of any conclusions in the study should be clearly detailed in the discussion. Overall a very interesting read and clearly thorough background research was done in preparing these models.

PLOS authors have the option to publish the peer review history of their article (what does this mean?). If published, this will include your full peer review and any attached files.

Reviewer #1: No

Reviewer #2: No

Reviewer #3: No
---

## [Decision Letter · Decision Letter 1]

17 Apr 2021

Dear Prof Torgerson,

Thank you very much for submitting your manuscript "Association between environmental and climatic risk factors and the spatial distribution of cystic and alveolar echinococcosis in Kyrgyzstan" for consideration at PLOS Neglected Tropical Diseases. As with all papers reviewed by the journal, your manuscript was reviewed by members of the editorial board and by several independent reviewers. The reviewers appreciated the attention to an important topic. Based on the reviews, we are likely to accept this manuscript for publication, providing that you modify the manuscript according to the review recommendations. 

Thank you for revising your manuscript in accordance with reviewer and editor concerns.

You'll see two reviewers re-reviewed the manuscript and were quite happy with the revisions.

I also re-reviewed the manuscript and I am happy with the edits made. Thank you for including table S2 to help the reader understand the identity of those correlated variables and the strength of the correlation between them and the main variable you included (10-year lag temperature). I am still dubious about the very strong correlations of 13 other temperature related temperature variables that were excluded due to collinearity with the main variable in the model 10-year time lagged mean temperature.

My concern is that there was no attempt to determine whether one or the other of those linked variables had a "best" fit with the AE cases. And the correlations are quite high (>.99 in most cases) so there may be no way to determine best fit with correlations that high anyways. Yet, it makes a big difference whether causally, one should expect higher AE cases 10 years after a particularly cool year or 5 years after, or 15 years after. The paper makes it seem that the 10-year lag has a very definitive meaning, when in fact, one might have found an equally good association with 5-year or 15-year lag temperature data. Or for that matter, with the temperature data in that year or the previous year (which wasn't tested).

I don't think this is a fatal flaw, but the issue is that you can definitively say that cooler temperatures, possibly with some time lag, may be associated with higher AE cases, but from this analysis you cannot say the exact time lag is definitively known, at least not from this dataset. This uncertainty should be addressed within the main body of the manuscript more explicitly, perhaps even as soon as in the abstract. OR spend a bit more time justifying the casual thinking, such as why one would include the 10 year time lagged temperature and not 5 or 15 year. Perhaps you might also point readers towards time series analyses that would be needed (or have already been done) to determine the appropriate time lag with which to evaluate the temperature data. A more in depth discussion of these points would be useful given that no other environmental variables emerged as significant in the analysis of AE or CE cases.

Therefore, I'm recommending one more minor revision to address this concern. You'll also see one minor suggestion from Reviewer #3 about line 177.

Sincerely,

Susanne H. Sokolow

Associate Editor

Justin Remais

Deputy Editor

Thank you for revising your manuscript in accordance with reviewer and editor concerns.

You'll see two reviewers re-reviewed the manuscript and were quite happy with the revisions.

I also re-reviewed the manuscript and I am happy with the edits made. Thank you for including table S2 to help the reader understand the identity of those correlated variables and the strength of the correlation between them and the main variable you included (10-year lag temperature). I am still dubious about the very strong correlations of 13 other temperature related temperature variables that were excluded due to collinearity with the main variable in the model 10-year time lagged mean temperature.

My concern is that there was no attempt to determine whether one or the other of those linked variables had a "best" fit with the AE cases. And the correlations are quite high (>.99 in most cases) so there may be no way to determine best fit with correlations that high anyways. Yet, it makes a big difference whether causally, one should expect higher AE cases 10 years after a particularly cool year or 5 years after, or 15 years after. The paper makes it seem that the 10-year lag has a very definitive meaning, when in fact, one might have found an equally good association with 5-year or 15-year lag temperature data. Or for that matter, with the temperature data in that year or the previous year (which wasn't tested).

I don't think this is a fatal flaw, but the issue is that you can definitively say that cooler temperatures, possibly with some time lag, may be associated with higher AE cases, but from this analysis you cannot say the exact time lag is definitively known, at least not from this dataset. This uncertainty should be addressed within the main body of the manuscript more explicitly, perhaps even as soon as in the abstract. OR spend a bit more time justifying the casual thinking, such as why one would include the 10 year time lagged temperature and not 5 or 15 year. Perhaps you might also point readers towards time series analyses that would be needed (or have already been done) to determine the appropriate time lag with which to evaluate the temperature data. A more in depth discussion of these points would be useful given that no other environmental variables emerged as significant in the analysis of AE or CE cases.

Therefore, I'm recommending one more minor revision to address this concern. You'll also see one minor suggestion from Reviewer #3 about line 177.

Reviewer's Responses to Questions

**Key Review Criteria Required for Acceptance?**

**Methods**

-Are the objectives of the study clearly articulated with a clear testable hypothesis stated?

-Is the study design appropriate to address the stated objectives?

-Is the population clearly described and appropriate for the hypothesis being tested?

-Is the sample size sufficient to ensure adequate power to address the hypothesis being tested?

-Were correct statistical analysis used to support conclusions?

-Are there concerns about ethical or regulatory requirements being met?

Reviewer #1: Yes

Reviewer #3: The objectives have been revised and are now clear and concise. The study design is thorough and appropriate to evaluate the stated objectives. Methodology has been revised to improve readability and contained within the methods section rather than in the results as well. Methods are a worthwhile contribution to the growing field of spatial epidemiology studying climatic factors on disease distribution.

**Results**

-Does the analysis presented match the analysis plan?

-Are the results clearly and completely presented?

-Are the figures (Tables, Images) of sufficient quality for clarity?

Reviewer #1: Perhaps, according to authors judgement based on other reviewers.

Reviewer #3: The analysis matches the analysis plan. The figure is very neat and attractive and is sufficient quality. The tables are organized and well-presented. Inclusion of supplemental tables adds value to the manuscript. Results section has been revised to present results more clearly. There is still some explaining of methods in results section, but it reads well and can be left as is per author choice. The organization and intermixing of results and methods is no longer a distractor from the work.

**Conclusions**

-Are the conclusions supported by the data presented?

-Are the limitations of analysis clearly described?

-Do the authors discuss how these data can be helpful to advance our understanding of the topic under study?

-Is public health relevance addressed?

Reviewer #1: Yes

Reviewer #3: The conclusions are supported by the presented data. The limitations of the study have been clearly, promptly, and adequately discussed in comparison to previous iteration. This was the major revision needed and it has been sufficiently addressed. The authors discuss that the data may advance our understanding of factors influencing CE and AE distribution. Although the limited geography and use of surgical only cases may limit its widespread application but it does add an interesting example of a spatial epi analysis. Public health relevance is addressed in intro and discussion.

**Editorial and Data Presentation Modifications?**

Reviewer #1: Not applicable

Reviewer #3: No editorial and/or data presentation modifications, other than one suggestion below:

Line 177 – no need to define multicollinearity, can remove “namely a correlation….more explanatory variables”

**Summary and General Comments**

Reviewer #1: Thank you for a great deal of improvement from the lest version. It is better from my end, but you will have to address concerns from other reviewers (if any) to have the manuscript accepted.

Reviewer #3: Overall an interesting and well-written manuscript. No further revisions are recommended for consideration of acceptance.

Figure Files:

Data Requirements:

Reproducibility:

References

---

## [Editor Report · Decision Letter 2]

20 May 2021

Dear Prof Torgerson,

We are pleased to inform you that your manuscript 'Association between environmental and climatic risk factors and the spatial distribution of cystic and alveolar echinococcosis in Kyrgyzstan' has been provisionally accepted for publication in PLOS Neglected Tropical Diseases.

Best regards,

Susanne H. Sokolow

Associate Editor

Justin Remais

Deputy Editor

Thank you for making thorough changes consistent with those requested by the reviewers and editor.

---

## [Editor Report · Acceptance letter]

16 Jun 2021

Dear Prof Torgerson,

We are delighted to inform you that your manuscript, "Association between environmental and climatic risk factors and the spatial distribution of cystic and alveolar echinococcosis in Kyrgyzstan," has been formally accepted for publication in PLOS Neglected Tropical Diseases.

Best regards,

Shaden Kamhawi

co-Editor-in-Chief

Paul Brindley

co-Editor-in-Chief
